# Benchmarking Empirical Privacy Protection for Adaptations of Large Language Models

## Abstract

Recent work has applied differential privacy (DP) to adapt large language models (LLMs) for sensitive applications, offering theoretical guarantees. However, its practical effectiveness remains unclear, partly due to LLM pretraining, where overlaps and interdependencies with adaptation data can undermine privacy despite DP efforts. To analyze this issue in practice, we investigate privacy risks under DP adaptations in LLMs using state-of-the-art attacks such as *robust membership inference* and *canary data extraction*. We benchmark these risks by systematically varying the adaptation data distribution, from exact overlaps with pretraining data, through in-distribution (IID) cases, to entirely out-of-distribution (OOD) examples. Additionally, we evaluate how different adaptation methods and different privacy regimes impact the vulnerability. Our results show that distribution shifts strongly influence privacy vulnerability: the closer the adaptation data is to the pretraining distribution, the higher the practical privacy risk at the same theoretical guarantee, even without direct data overlap. We find that parameter-efficient fine-tuning methods, such as LoRA, achieve the highest empirical privacy protection for OOD data. Our benchmark identifies key factors for achieving practical privacy in DP LLM adaptation, providing actionable insights for deploying customized models in sensitive settings. Looking forward, we propose a structured framework for *holistic* privacy assessment beyond adaptation privacy, to identify and evaluate risks across LLMs' full pretrain-adapt pipeline.

## 1 Introduction

The use of *pretrained* large language models (LLMs) for sensitive downstream tasks, such as medical decision making, has grown rapidly [25, 12, 49]. To offer protection for the private data used to *adapt* the LLMs to these sensitive tasks, differential privacy (DP) [16, 17] has emerged as a gold standard [53, 54, 30, 13, 33]. However, adapting a pretrained LLM with DP may not always provide the anticipated privacy protections [48]. The challenge arises from potential overlap or complex interdependencies between data used to pretrain the LLMs and the adaptation dataset. The problem is exacerbated by the fact that for most LLMs, their pretraining datasets are not disclosed [35, 39, 46], rendering a structured reasoning of the interdependencies with the private adaptation data impossible.

While prior work has investigated privacy risks stemming from LLM pretraining [10, 9], post-hoc leakage in non-private adaptations [58], or auditing DP adaptations via synthetic canaries [36], we still lack a structured understanding of the *empirical privacy risks* of DP adaptations. This is a critical gap. Without a clear understanding of the practical risks, LLM practitioners are left with little guidance on how to privately apply LLMs in privacy-sensitive settings, including critical questions like: which adaptation method to use, what pretrained model is best given the private adaptation data distribution, and what privacy levels will be protective enough.

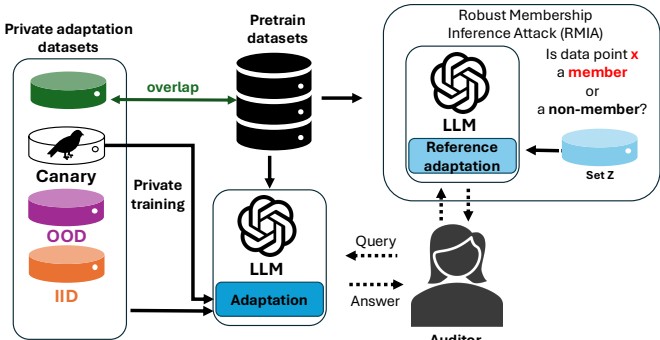

Figure 1: **Setup for Privacy Auditing of DP-LLM Adaptations.** We perform our audits based on the privately adapted LLM's output, either by using RMIA [8] as the strongest state-of-the-art membership inference attack, or by relying on data extraction attacks. For the latter, we include *canary* data into the adaptation set and measure its exposure.

To close this gap, we conduct a comprehensive benchmark evaluation that sheds light on the empirical leakage introduced by DP adaptations. We evaluate a wide range of private adaptation strategies, including full and last-layer DP fine-tuning [30], parameter-efficient fine-tuning (PEFT) methods such as DP-LoRA [21, 54], DP-Prefix Tuning [31], as well as DP prompting schemes [13]. To assess leakage, we focus on the *Robust Membership Inference Attack* (RMIA) [56], which represents the strongest state-of-the-art threat model for auditing LLM privacy, and complement this with *data extraction attacks* [47, 7, 6] to evaluate more severe forms of information leakage. A general overview of privacy auditing for adapted LLMs is provided in Figure 1.

We systematically analyze a spectrum of possible distributions for the adaptation data with respect to the pretraining data—ranging from data perfectly overlapping with the pretraining data, over IID scenarios, to entirely OOD examples—to understand the possible privacy implications for all setups. Our benchmark spans *six* datasets drawn from diverse domains, *four* adaptation methods, *and* six pretrained LLMs of different sizes and architectures, enabling comprehensive comparisons across setups. We further analyze a broad spectrum of privacy regimes from no privacy to high privacy, enabling structured reasoning about the resulting risks. Our study is guided by a central question: *What are the empirical privacy risks for the adaptation data that result from DP adaptations?*

Looking ahead, we highlight the need to jointly audit privacy risks from pretraining and adaptation and their interplay, as LLMs may leak information from either stage. To address this, we propose a new structured framework for holistic privacy assessment across the full pretrain-adapt pipeline. It defines four key audit stages: (1) pretraining, (2) adaptation, (3) their joint interaction, and (4) post-adaptation auditing of pretraining. To formally ground these audits and make them instantiatable, we redefine each stage's membership inference game [52, 23]. We hope this formalization and our practical insights from the benchmark will guide researchers in developing future assessments and help practitioners deploy customized LLMs responsibly in sensitive domains.

## 2 Background and Related Work

**Differential Privacy.** The mathematical framework of DP [16] formalizes the intuition that privacy guarantees can be obtained when a randomized mechanism $\mathcal{M}$ executed on two neighboring datasets $D$, $D'$ that differ in only one data point, yields roughly the same result, *i.e.,*

$$\Pr[\mathcal{M}(D) \in S] \leq e^\epsilon \cdot \Pr[\mathcal{M}(D') \in S] + \delta. \tag{1}$$

The privacy parameter $\varepsilon$ specifies how much the result can differ, and $\delta$ is the probability of failure to meet that guarantee. There are two canonical algorithms to implement DP guarantees in machine learning (ML): DPSGD (*Differentially Private Stochastic Gradient Descent*) algorithm [2], which extends standard stochastic gradient descent with clipping and noising gradients, and PATE (*Private Aggregation of Teacher Ensembles*) [37, 38], which is an inference time algorithm that privately transfers knowledge from an ensemble of teachers to a public student model.

**Private Adaptations of LLMs.** LLMs are pretrained on extensive amounts of public data, followed by adaptations to private downstream tasks. The existing methods for private LLM adaptations fall into two categories: (1) *private tuning methods*, such as PrivateLoRA [54] or PromptDPSGD [13], that rely on access to the LLM gradients and are based on the DPSGD algorithm, and (2) *private in-context learning (ICL) methods*, such as DP-ICL [51] or PromptPATE [13], which require only API (black-box) access to the LLM and are based on PATE. See Appendix A.1 for details.

**Membership Inference Attacks.** A membership inference attack (MIA) [44, 56, 43, 8] aims to determine whether a specific data point can be identified as part of a model's training set. This approach plays a crucial role in applications ranging from privacy assurance [45] to identifying protected or copyrighted content embedded in pretraining data [41]. While most MIA research has focused on supervised learning settings [8], new advancements reveal their broader relevance. Duan et al. [14] revealed a discrete-prompt-based MIA, disclosing vulnerabilities in proprietary LLMs like GPT-3, which risk leaking private information through prompt-based queries [13]. See Appendix A.2 for an in-depth discussion of the existing attacks.

**Canary Exposure and Data Extraction Attacks.** An alternative to membership inference attacks (MIAs) for evaluating privacy leakage in machine learning models is to measure the *exposure* of training data. Given a universe of candidates $\mathcal{U}$ and an attacker's ranking $\hat{Z}$ by likelihood of membership, the exposure of a target sample $z \in \mathcal{U}$ is defined as:

$$\textbf{exposure}(z, \hat{Z}) = \log_2 |\mathcal{U}| - \log_2\big(\text{rank}(z; \hat{Z})\big). \qquad (2)$$

This score is maximal when $z$ is ranked most likely and zero when ranked least likely. In a complementary vein, *extractability* quantifies how readily a model emits a secret string when prompted. A suffix $s$ is said to be *extractable with $k$ tokens of context* if there exists some prefix $p$ of length $k$ such that, under greedy decoding, the model outputs $s$ immediately following $p$. When $s$ is sufficiently long and random, its extractability serves as a practical metric of memorization in LLMs. Further discussion appears in Appendix A.3.

**Benchmarking Privacy Vulnerabilities.** Zhu et al. [58] introduced *PrivAuditor*, which systematically and empirically evaluates the privacy leakage from LLM adaptations. In contrast to our work, they focus on *non-private* adaptations only. Li et al. [27] evaluated the privacy leakage of private LLMs adaptations through empirical privacy attacks, such as data extraction, MIAs, and embedding-level privacy attacks. This benchmark focuses mostly on tradeoffs between privacy and utility, highlighting the complexity of balancing them. Contrary to our work, this work does not explore the relationship between the pretraining data and the fine-tuning one. *LLM-PBE* [28] empirically evaluates privacy risks throughout the LLM lifecycle, including pretraining, fine-tuning, and querying. Zhou et al. [57] investigated potential data leakage across widely used software engineering benchmarks.

## 3 Experimental Setup

We begin by detailing the setup used for our benchmark. Further details are presented in Appendix B.

**Models and Pretraining Data.** Our work primarily focuses on the Pythia family of models trained on the Pile dataset [18], and the GPT-Neo family [4]. To benchmark the effects over various model sizes, we use Pythia 1.4B, Pythia 1B, Pythia 410M, Pythia 160M, Pythia 70M, GPT Neo 1.3B, and GPT Neo 125 M. The Pile dataset [18] is an 800GB collection of diverse English-language datasets, including text from sources such as books, academic papers, or source code repositories. In all cases where a specific model is not explicitly mentioned, we use Pythia 1B as the default model.

**Adaptation Datasets.** We categorize the datasets used in our experiments into **in-distribution (IID)** and **out-of-distribution (OOD)**, depending on their relationship to the pretraining data. IID datasets come from the same distribution as the pretraining data, and we identify two cases: one with a full overlap between pretraining and adaptation data, where we use data directly from the pretraining set for the adaptations, and one with no overlap, where the data is sourced from the corresponding validation set from the pretraining distribution. We focus on the following Pile subsets for the IID datasets: BookCorpus2, GitHub, and Enron Emails [24]. In contrast, OOD datasets are derived from a different distribution and do not overlap with pretraining data. Thereby, we choose SAMSum [19], and GermanWiki [1]. We elaborate more in Appendix B.1.

**Adaptation Methods.** We evaluate different types of adaptations, including fine-tuning of all model parameters [30], or the last layer (*i.e.,* the head) and PEFT methods, such as LoRA [21, 54] and Prefix

Tuning [31, 13]. Considering a Pythia 1B model, we train 1B parameters for Full Fine-Tuning, 1M for LoRA, 130M for Prefix Tuning, and 100M for last-layer (Head) Fine-Tuning. Since membership inference success is highly dependent on the train-test gap, for a fair comparison of the privacy leakage, we ensure similar evaluation perplexities, in particular, similar validation loss values at the end of the adaptation's training for specific datasets across adaptation methods, see Appendix B.2.

**Membership Inference.** For membership inference, we rely on the strongest state-of-the-art attack, namely RMIA (Robust Membership Inference Attack) [56]. We use its offline version because it is computationally effective and does not require training customized reference models for each targeted sample (as in the online version of the attack). We also leverage a single reference model for our experiments, as the authors show strong MIA performance even with a single reference model. We consider different types of reference models. Unless explicitly stated, we focus on using a "shadow" model (adaptation), in our case Pythia 1B, which is trained in the same way as the target model, but on a different split of the same fine-tuning data. We also evaluate the *Reference* method [7], which calibrates the target model's loss using a reference model, and compare against Min-K% as a reference-less baseline attack. As with RMIA, we report the best AUC from a grid search over Min-K%'s parameter $K$. See Appendix B.4 for a detailed description of the setup.

**Canary Exposure and Data Extraction Attacks.** To evaluate memorization, we insert adversarial canaries into a small portion of the adaptation data and estimate their exposure using two approximation methods: sampling and distribution modeling. Both approaches perform similarly when using 256 non-member canaries, and we adopt sampling for efficiency. Moreover, when considering $k$-extractable memorization, we set $k = 10$ tokens. A detailed description of the data extraction setup is provided in Appendix B.5.

## 4 Benchmark design and experiments

To address our benchmark's central question—*What are the empirical privacy risks to adaptation data under DP adaptations?*—we break it down into five concrete research questions.

### 4.1 RQ1: How does the relationship (overlapping, IID, OOD) between adaptation and pretraining datasets impact data privacy?

**Motivation.** The pretrain-adapt paradigm uses LLMs pretrained on large public datasets, which are then adapted to smaller, often sensitive, private datasets using DP methods. While DP offers formal guarantees, its practical effectiveness under the pretrain-adapt paradigm remains unclear—particularly how the relationship and interplay between adaptation and pretraining data (*e.g.,* overlapping, IID, or OOD) influences actual privacy leakage.

**Summary of Findings.** Our results show that (1) privacy risks increase when the adaptation data distribution is closer to the pretraining data, even if there is no direct overlap. (2) Surprisingly, IID data from the pretraining validation set leaks as much as directly overlapping data, underscoring distributional closeness as the main driver of risk.

**Detailed Results.** We present our main results in Table 1 and Table 2. We focus our discussion on Pythia-1B, and further expand it for the other models in Appendix C.1. They show that the average AUC is generally higher in IID settings than OOD in all attacks and adaptations. For instance, looking at *RMIA (shadow)* using $\varepsilon = 8$, we observe that the average AUC is between 0.7 and 0.9 in the IID setting, while it is between 0.63 and 0.87 for the OOD setting. More detailed analyses for different attack setups and more privacy regimes are depicted in Appendix C.1. We also identify distributional closeness as a key risk factor, as overlapping data leaks similarly to IID. Moreover, our results indicate that under both a strong attack and in more practical scenarios, moderate privacy regimes (*e.g.,* $\varepsilon = 8$) still present a real threat of privacy leakage from IID. On the other hand, under this regime, privacy leakage from the OOD is mostly observed with a strong attack. Moreover, in Appendix C.4, Figure 8 shows over the training epochs the Overlap (Train) and IID data (Val) privacy leakage, and further highlights a similar privacy leakage between Overlap and IID data across the whole training run. We also analyze the impact of subset characteristics on privacy leakage in Appendix C.3, and we discover that the pretraining dataset size and complexity influence the privacy leakage in the training datasets. We observe that privacy leakage increases with both the size and complexity of the subsets. Larger datasets produce more IID results than smaller subsets, further validating our findings.

Table 1: **Membership Inference for OOD Adaptations.** We audit only the adaptations and assume the same pretrained LLM is used for all adaptations. We present the AUC scores obtained with RMIA MIAs for the Pythia 1B model adapted on different datasets with $\varepsilon \in \{0.1, 8, \infty\}$.

| MIA | Dataset / Adaptation | SAMSum | | | GermanWiki | | | Average | | |
|---|---|---|---|---|---|---|---|---|---|---|
| | | $\varepsilon = \infty$ | $\varepsilon = 8$ | $\varepsilon = 0.1$ | $\varepsilon = \infty$ | $\varepsilon = 8$ | $\varepsilon = 0.1$ | $\varepsilon = \infty$ | $\varepsilon = 8$ | $\varepsilon = 0.1$ |
| RMIA (shadow) | Prefix Tuning | 1.00 | 0.62 | 0.63 | 1.00 | 0.64 | 0.61 | 1.00 | 0.63 | 0.62 |
| | LoRA | 0.86 | 0.69 | 0.50 | 1.00 | 0.59 | 0.66 | 0.93 | 0.64 | 0.58 |
| | Full Fine-Tune | 1.00 | 0.82 | 0.62 | 1.00 | 0.71 | 0.55 | 1.00 | 0.77 | 0.59 |
| | Head Fine-Tune | 1.00 | 0.98 | 0.62 | 1.00 | 0.76 | 0.70 | 1.00 | 0.87 | 0.66 |
| | Average | 0.97 | 0.78 | 0.59 | 1.00 | 0.67 | 0.63 | 0.98 | 0.73 | 0.61 |
| Reference (Pythia 1B) | Prefix Tuning | 0.93 | 0.50 | 0.51 | 0.92 | 0.50 | 0.50 | 0.92 | 0.50 | 0.50 |
| | LoRA | 0.51 | 0.51 | 0.51 | 0.82 | 0.51 | 0.51 | 0.66 | 0.51 | 0.51 |
| | Full Fine-Tune | 0.94 | 0.51 | 0.51 | 0.99 | 0.51 | 0.50 | 0.96 | 0.51 | 0.51 |
| | Head Fine-Tune | 0.97 | 0.52 | 0.51 | 0.98 | 0.51 | 0.50 | 0.97 | 0.51 | 0.50 |
| | Average | 0.84 | 0.51 | 0.51 | 0.93 | 0.51 | 0.50 | 0.88 | 0.51 | 0.51 |

Table 2: **Membership Inference for in-distribution (IID) Adaptations** using the setup from Table 1.

| MIA | Dataset / Adaptation | Bookcorpus2 Val | | | Bookcorpus2 Train | | | Github Val | | | Enron Val | | | Average | | |
|---|---|---|---|---|---|---|---|---|---|---|---|---|---|---|---|---|
| | | $\varepsilon = \infty$ | $\varepsilon = 8$ | $\varepsilon = 0.1$ | $\varepsilon = \infty$ | $\varepsilon = 8$ | $\varepsilon = 0.1$ | $\varepsilon = \infty$ | $\varepsilon = 8$ | $\varepsilon = 0.1$ | $\varepsilon = \infty$ | $\varepsilon = 8$ | $\varepsilon = 0.1$ | $\varepsilon = \infty$ | $\varepsilon = 8$ | $\varepsilon = 0.1$ |
| RMIA (shadow) | Prefix Tuning | 1.00 | 0.89 | 0.56 | 1.00 | 0.90 | 0.55 | 1.00 | 0.93 | 0.63 | 1.00 | 0.88 | 0.58 | 1.00 | 0.90 | 0.58 |
| | LoRA | 1.00 | 0.70 | 0.52 | 1.00 | 0.69 | 0.53 | 1.00 | 0.74 | 0.52 | 1.00 | 0.73 | 0.52 | 1.00 | 0.71 | 0.52 |
| | Full Fine-Tune | 1.00 | 0.75 | 0.77 | 1.00 | 0.75 | 0.76 | 1.00 | 0.78 | 0.80 | 1.00 | 0.91 | 0.66 | 1.00 | 0.80 | 0.75 |
| | Head Fine-Tune | 1.00 | 0.72 | 0.73 | 1.00 | 0.72 | 0.72 | 1.00 | 0.80 | 0.74 | 1.00 | 0.57 | 0.65 | 1.00 | 0.70 | 0.71 |
| | Average | 1.00 | 0.77 | 0.65 | 1.00 | 0.76 | 0.64 | 1.00 | 0.81 | 0.67 | 1.00 | 0.77 | 0.60 | 1.00 | 0.78 | 0.64 |
| Reference (Pythia 1B) | Prefix Tuning | 0.93 | 0.56 | 0.52 | 0.97 | 0.57 | 0.50 | 0.97 | 0.53 | 0.51 | 0.97 | 0.54 | 0.50 | 0.96 | 0.55 | 0.51 |
| | LoRA | 0.89 | 0.52 | 0.52 | 0.97 | 0.51 | 0.50 | 0.92 | 0.51 | 0.50 | 0.97 | 0.55 | 0.51 | 0.94 | 0.52 | 0.51 |
| | Full Fine-Tune | 1.00 | 0.54 | 0.52 | 1.00 | 0.54 | 0.52 | 0.99 | 0.54 | 0.52 | 0.98 | 0.59 | 0.50 | 0.99 | 0.55 | 0.51 |
| | Head Fine-Tune | 0.98 | 0.57 | 0.52 | 1.00 | 0.56 | 0.51 | 0.99 | 0.66 | 0.50 | 0.99 | 0.54 | 0.50 | 0.99 | 0.58 | 0.51 |
| | Average | 0.95 | 0.55 | 0.52 | 0.98 | 0.55 | 0.51 | 0.97 | 0.56 | 0.51 | 0.98 | 0.55 | 0.50 | 0.97 | 0.55 | 0.51 |

## 4.2 RQ2: Which DP adaptation method is the most protective?

**Motivation.** It is known that the type of adaptation has a significant impact on the utility of the final model [58]. However, different adaptations might also offer disparate empirical protection at the same formal privacy guarantee, motivating our empirical comparison.

**Summary of Findings.** While LoRA provides much better empirical privacy protection in non-private settings compared to other adaptations, the differences become more subtle under the DP regime. Despite this, LoRA consistently achieves a relatively low AUC, whereas the other adaptations show varying trends depending on the dataset or privacy budget.

**Detailed Results.** Specifically, as shown in Table 1 for OOD datasets with $\varepsilon = 8$, the most vulnerable adaptations are Full and Head Fine-Tune. On the other hand, for IID data, the strongest protection provides Head Fine-Tune, which is marginally better LoRA. With stronger privacy guarantees, LoRA is the most private for OOD datasets with an AUC score of 0.58, thus slightly better than Full Fine-Tune. On the other hand, while adapting to the IID dataset, LoRA outperforms other adaptations. Notably, Full Fine-Tune and Head Fine-Tune show much lower privacy protection in these settings.

## 4.3 RQ3: Are the same adaptations robust against data extraction?

**Motivation.** Data extraction attacks are even more severe than MIAs. Therefore, it is crucial to evaluate the protectiveness of DP adaptations against this stronger threat.

**Summary of Findings.** We find that Prefix Tuning is the most vulnerable adaptation method in this setting. On the other hand, LoRA and Head Fine-Tune in both cases, with and without DP guarantees exhibit resistance against data extraction.

**Detailed Results.** We report detailed results in Appendix C.2. In particular, Table 17 and Table 18 show that for $\varepsilon = 0.1$ the exposure is around 1.44, therefore, close to random guessing. We also noticed a limited influence on the choice of the canary prefix type. Moreover, the adversarial prefix is the main source of privacy leaks, with the interaction between the prefix and the individual sample playing a smaller role, see Figure 9 in Appendix C.5.

## 4.4 RQ4: How important is the attacker's knowledge of the pretrained model?

**Motivation.** The attacker's knowledge of the pretrained model plays a crucial role in the success of MIAs, as it enables them to select more relevant reference models and non-member data for training,

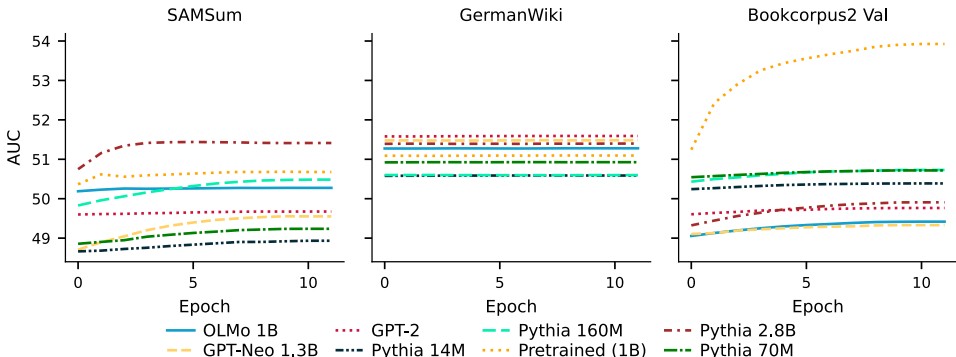

Figure 2: **IID data is more susceptible to leakage using the pretrained base model than OOD data.** We compare the effectiveness of performing RMIA on fully fine-tuned Pythia 1B with $\varepsilon = 8$ with different pretrained models as reference models.

which is one of the main challenges of MIAs [50, 8]. We investigate various setups, including an attacker who has access to a shadow model from the same pretraining distribution as the adapted LLM, a similar model, and no access to external models. This helps us characterize the landscape of potential real-world risks and setups.

**Summary of Findings.** MIAs' performance highly depends on the attacker's knowledge of the target model and pretraining data. In particular, RMIA performs best when a shadow model shares architecture, initialization weights, and training data distribution. Meanwhile, MIAs' effectiveness rapidly deteriorates as shadow models are trained on different distributions or architectures. Particularly, we observe that when a shadow model trained on the same distribution of the target model is available, using the pretrained model is the second-best choice, followed by models of the same family and similar size.

**Detailed Results.** To simulate attackers with various background knowledge, in this setting, we also consider other "shadow" models: Pythia 14M, Pythia 160M, Pythia 1B, Pythia 2.8B [3], GPT-neox [4], OLMo-1B [20], and GPT-2 [40]. The MIA performance is close to random for private adaptations with $\varepsilon = 8$. Furthermore, as shown in Figure 2, while the MIA's performance for Pythia 1B is higher on IID data, the choice of reference model has little effect when attacking models adapted on OOD data, even with architectural differences between the model and the reference model *i.e.,* GPT-Neo 1.3B and OLMo 1B. Moreover, as in the other case, Figure 11 (in Appendix D) shows that the privacy leakage is similar between IID and the corresponding overlapping data. We show further experiments in Appendix D.

### 4.5 RQ5: How does adaptation change the pretraining dataset vulnerability?

**Motivation.** DP adaptations only guarantee protection for the adaptation dataset. Yet, adapting the model to other data, while introducing noise, can also affect the pretraining leakage. This is an important aspect to study, as also pretraining data can be private [48], *e.g.,* private conversations with ChatGPT used to improve the models, or emails used to pretrain Gemini. Therefore, we also empirically investigate how adapting pretrained LLMs affects the leakage of pretraining data.

**Summary of Findings.** Our findings show that the choice of adaptation method impacts the privacy of pretraining data. Specifically, our evaluation shows that Prefix Tuning reduces the leakage of memorized pretraining data from adapted language models, especially in high-privacy settings. However, for the other adaptations, this effect is negligible, and the adapted model retain most of the pretraining memorization.

**Detailed Results.** We evaluate the effect of OOD and IID adaptation data on the leakage of memorized pretraining data from the adapted LLM. Specifically, as we show in Figure 3, Prefix Tuning significantly reduces leakage, particularly in high-privacy regimes. For the other adaptation methods, the number of memorized samples often remains above 460 samples. For Prefix Tuning, the number of memorized samples is often lower than 460 and goes down to around 430 with $\varepsilon = 0.1$, thus suggesting that adaptation partially mitigates the pretraining memorization.

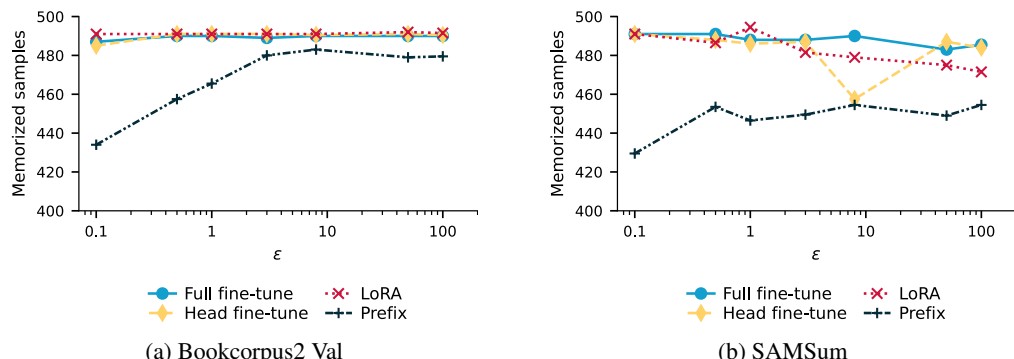

(a) Bookcorpus2 Val        (b) SAMSum

Figure 3: **Fewer memorized samples after prefix tuning.** There are fewer verbatim generations of training samples after the prefix tuning, especially for small $\varepsilon$ values. We present the number of memorized samples from the Pile that remain memorized after adapting Pythia 1B on Bookcorpus2 val and SAMSum datasets. The evaluation was done for $\varepsilon = \{0.1, 1, 3, 8, 50, 100, \infty\}$. We present the x-axis using a log scale.

## 5   Discussion of our Results

Our findings reveal a complex interplay between pretraining and adaptation data. This significantly affects the privacy risks under DP adaptations. Below, we discuss the implications of these findings when adapting pretrained LLMs to sensitive domains using DP.

**Disparate Leakage Based on Distribution.** Our results demonstrate that the distributional closeness between pretraining and adaptation data is a key factor influencing empirical privacy leakage under DP. Adaptations using IID data—data from the same distribution but not seen during pretraining— consistently showed the highest vulnerability. This presents a fundamental trade-off: while adapting a model already pretrained on similar data is often beneficial for utility, it simultaneously increases privacy risk.

**Disparate Leakage Based on Adaptation Method.** We also observe that not all DP adaptation methods offer equal protection, even when enforcing the same formal level guarantee, expressed in the same $\varepsilon$. This aligns with earlier findings in the non-private regime, where privacy-utility trade-offs differ across methods [58]. In our experiments, LoRA appeared most consistently robust against privacy attacks, while Prefix Tuning showed the least vulnerability to extraction attacks. These differences are highly relevant for practice: in addition to choosing methods that optimize downstream performance, practitioners should also consider empirical privacy leakage. The attacks we use in this paper offer a way to assess and understand such risks under realistic conditions.

**Choosing a Privacy Regime.** We find that in moderate privacy regimes, *e.g.,* $\varepsilon = 8$, sensitive adaptation data still experiences significant practical vulnerability against both MIAs and data extraction attacks. This highlights the necessity to perform private LLM adaptations in the high-privacy regime, *i.e.,* with low $\varepsilon$ to achieve practical protection.

**Reliance on Accurate Shadow Model.** We show that attackers gain a substantial advantage when they have access to the original pretrained LLM used during adaptation. Shadow models instantiated with the same pretrained model as the adapted LLM's base consistently achieved higher attack success. This is especially concerning given the rise of adapting publicly available LLMs, which makes strong shadow models easily accessible to adversaries. These findings further underscore the need for stringent privacy settings in DP adaptations.

**Towards a Holistic Privacy Auditing for LLMs** Our results suggest that privacy assessments should not treat pretraining and adaptation in isolation. The strong interdependence between these stages demands holistic analysis. Motivated by this insight, we introduce a structured framework in the next section that formalizes how privacy assessments and audits under the pretrain-adapt paradigm should be conducted. We hope this framework encourages the development of privacy assessment methods that match the complexity of modern private LLM pipelines.

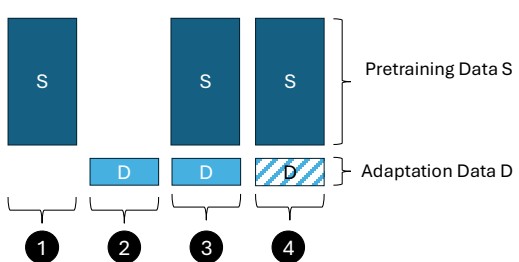

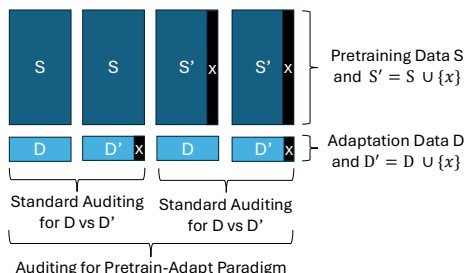

Figure 4: **Stages of Auditing.** We analyze four stages of auditing: ❶ Audit Pretraining, ❷ Audit Adaptations, ❸ Joint Auditing of Pretraining and Adaptations, ❹ Post-Adaptation Auditing of the Pretraining.

Figure 5: **Setup for Joint Adaptation auditing (3).** We consider different datasets for pretraining and adaptation, distinguishing it from standard ML privacy auditing [34, 55] by considering pretraining data.

# 6 Towards Holistic Privacy Audits under the Pretrain-Adapt Learning Paradigm

## 6.1 From Stages to Adversary Game under Pretrain-Adapt Privacy Auditing

While our understanding of empirical privacy risks has grown, we recognize the need to go further and adopt more nuanced approaches to tackle privacy risks posed during the adaptation of LLMs. Therefore, we formalize a framework to assess privacy risks holistically for LLMs and their pretrain-adapt paradigm. In total, we identify four different stages of auditing that need to be considered (see Figure 4) under the pretrain-adapt paradigm, namely (1) audit pretraining, (2) audit adaptations, (3) joint audit of pretraining and adaptations, and (4) post-adaptation auditing of the pretraining, as shown in Figure 4. Based on them, we formalize how to instantiate these audits and contrast them with standard privacy auditing. Privacy audits can be modeled as an *adversarial game* $\mathcal{G}$ [52, 23] where the main task is to guess if a given data point $x$ was in a model's training set or not. This game can, therefore, also be referred to as the *membership inference game*. We define the adversarial game $\mathcal{G}$ analogous to the one for standard ML, yet take two datasets, $S$ the pretraining data, and $D$ the adaptation data into account. Additionally, we denote the pretraining procedure by $T$ and the adaptation procedure by $T'$. We mark the deviations to the original game in blue.

1. The challenger samples $a \xleftarrow{\text{R}} \{0, 1\}$ and $b \xleftarrow{\text{R}} \{0, 1\}$ (where $a$ and $b$ are binary variables)

2. The challenger trains a model $\theta \xleftarrow{\text{T}} \tilde{S}, \theta_0$, where $\tilde{S} = S$ if $a = 0$, otherwise $\tilde{S} = S \cup \{x\}$

3. The challenger adapts $\theta$ such that $\theta' \xleftarrow{\text{T'}} \tilde{D}$, where $\tilde{D} = D$ if $b = 0$, otherwise $\tilde{D} = D \cup \{x\}$

4. The challenger sends $\theta'$ to the attacker

5. The attacker guesses $\hat{a}, \hat{b} \leftarrow \mathcal{A}(\theta, \theta', x)$

Whether the attacker has to guess both $\hat{a}, \hat{b}$ and what background knowledge they have, *i.e.,* whether they get access to both $\theta$ and $\theta'$ depends on the auditing stage. We detail the attacker's background knowledge and guesses—formulated as hypotheses with a null hypothesis $H_0$ and an alternative hypothesis $H_A$—for the four auditing stages from our taxonomy.

**(1) Auditing pretraining** resembles standard ML auditing, targeting privacy leakage from pretrained models. Differences arise from larger datasets and models, limiting both DP protection efficacy [10] and applicability of auditing techniques like MIA [15]. In this setting, the challenger releases the pretrained model $\theta$ to the attacker. The attacker's goal is to correctly guess whether $x$ was in the pretraining data $S$. Their guesses $\hat{a}$, are over the random variable $a$.

$$H_0 : a = 0 \qquad H_A : a = 1$$

**(2) Auditing adaptation** a new pretrain-adapt paradigm aspect, detects adaptation dataset leakage from adapted LLMs. The key differentiating factor of privacy audits in standard ML is using a

pretrained model that the adaptations are trained on instead of a random initialization. We assume the same pretrained model is used for all the considered adaptations in an adaptation audit. In this setting, the challenger releases only the adapted model $\theta'$ to the attacker. The attacker does not know whether $x \in S$ or not and considers only the adaptation. Their guesses $\hat{b}$, are, hence, over the random variable $b$.

$$H_0 : b = 0 \qquad H_A : b = 1$$

**(3) Joint auditing** evaluates combined leakage from both pretraining and adaptation datasets in the adapted LLM. Typical privacy preservation involves non-DP-trained LLMs with DP-trained adaptations. In this setting, the challenger releases both the pretrained model $\theta$ and the adapted $\theta'$ to the attacker. Depending on the attacker's background knowledge, we consider three possible cases

| The attacker knows that $x \notin S$ and guesses $b$. | The attacker knows that $x \in S$ and guesses $b$. | The attacker knows that the target sample $x$ is either in both (pretraining and adaptation sets) or neither of them and guesses $(a, b)$. |
|---|---|---|
| $H_0 : (a, b) = (0, 0) \quad H_A : (a, b) = (0, 1)$ | $H_0 : (a, b) = (1, 0) \quad H_A : (a, b) = (1, 1)$ | $H_0 : (a, b) = (0, 0) \quad H_A : (a, b) = (1, 1)$ |

**(4) Post-Adaptation Auditing** evaluates how the (private) adaptations influence the potential protection of the data points used for pretraining, which is usually conducted without any formal guarantees. Changes to the model behavior induced through adaptations or noise added during their training might influence the effective exposure of pretraining data from model predictions. In this setting, the challenger releases both the pretrained $\theta$ and the adapted $\theta'$. It is known that the target sample $x$ is not in $D$ and the attacker guesses $a$.

$$H_0 : (a, b) = (0, 0) \qquad H_A : (a, b) = (1, 0)$$

In essence, auditing pretraining considers only the pretraining itself. Similarly, auditing the adaptations considers the adaptations themselves. On the other hand, the joint adaptation reasons about both pretraining and adaptation sets. Finally, the post-adaptation auditing is only for the pretraining set, but the applied adaptation influences the auditing.

## 6.2 Practical Application of Holistic Audits

Our new perspective on the pretrain-adapt paradigm gives both practitioners and researchers clearer insights into each threat model's risks. Formalizing the auditing setup supports systematic reasoning about privacy risks, thus clarifying the guarantees that different methods need to provide. Therefore, our formalization allows for creating a unified interface for measuring privacy leakage, regardless of whether its source is pretraining or adaptation data. Moreover, our work demonstrates that looking at pretraining and adaptation components separately can lead to a false impression of privacy. The connection between these stages affects privacy leakage, which makes comprehensive auditing essential within pretrain-adapt paradigm. We believe that developing and sharing tools that support all privacy assessment stages, from threat modeling and risk quantification to mitigation, will empower the research community to more effectively define risks and allow for the reduction of privacy risks in practice.

## 7 Conclusions

In this work, we benchmark the practical privacy risks that arise under DP adaptations of LLMs within the pretrain-adapt paradigm. Our comprehensive empirical analysis confirms the theoretical concern that pretraining significantly amplifies the privacy risks associated with the *adaptation data*. We find that the closeness of adaptation and pretraining data distributions plays a critical role: even in the absence of overlap, higher distributional similarity results in increased privacy leakage. Additionally, we observe that the choice of adaptation method impacts privacy leakage, with PEFT methods, such as LoRA, offering significantly lower privacy risks while maintaining strong utility. Furthermore, we show Prefix Tuning can reduce the leakage of pretraining data, likely due to the added input noise during private adaptation. Our findings highlight the need for stringent DP constraints (*e.g.,* $\varepsilon < 0.1$) to mitigate privacy risks in LLM adaptations effectively. It also motivates the need for holistic privacy assessments under the pretrain-adapt paradigm and takes the first step towards it by formalizing such an assessment over the different stages. This work lays a foundational framework for future research efforts aimed at safeguarding privacy within the pretrain-adapt paradigm.

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
