# OpenReview forum: "Benchmarking Empirical Privacy Protection for Adaptations of Large Language Models"
_NeurIPS.cc/2025/Datasets_and_Benchmarks_Track — Submitted to NeurIPS 2025 Datasets and Benchmarks Track_

### Official Review · Reviewer_MdW3 · 2025-07-03

**Rating:** 5
**Confidence:** 2

**Summary:**

This work presents a comprehensive benchmark evaluating the practical privacy risks in differentially private adaptations of LLMs. The authors systematically investigate how the relationship between pretraining and adaptation data distributions affects privacy leakage, examining scenarios from exact overlap to out-of-distribution cases. They employ attack methods including RMIA and data extraction across multiple adaptation techniques (LoRA, Prefix Tuning, full fine-tuning) and propose a framework for privacy auditing in the pretrain-adapt paradigm. The work spans six datasets, four adaptation methods, and six LLMs of varying sizes.

**Dataset Code Accessibility:**

Yes

**Ethical Considerations:**

No, there are no or only very minor ethics concerns

**Final Justification:**

My questions are adequately answered, and I appreciate the authors for offering the off-the-shelf tool in their new pipeline. Therefore, I will keep my current rating.

I also read through other reviewers' comments, but found myself not very familiar with this area and had some bias. Therefore, I have lowered my confidence to be responsible.

**Limitations Weaknesses:**

1. Including theoretical analysis of why certain adaptation methods work would be great

2. Including error bar could make the empirical results and conclusions more convincing

3. The evaluation focuses primarily on the Pythia family and GPT-Neo models. Including more diverse architectures would be great (e.g. gemma).

**Strengths Contributions:**

1. This paper is overall well-written with clear structure, informative figures and tables

2. The benchmark demonstrates exceptional thoroughness, systematically varying critical factors including data distribution relationships (overlap, IID, OOD), adaptation methods, privacy regimes, and model architectures.

3. The work addresses a critical gap between theoretical DP guarantees and empirical privacy protection in real-world LLM deployment scenarios.

---

> ### Author Rebuttal · Authors · 2025-07-31
>
> We thank the Reviewer for their positive feedback and for recognizing our benchmark as “exceptionally thorough”, systematic, and bridging the critical gap for real-world LLMs deployments.
>
> >**Including a theoretical analysis of why certain adaptation methods work would be great**
>
> We agree that theoretical insights into why certain adaptation methods offer stronger empirical privacy would be valuable. However, our focus in this paper was to establish a systematic empirical benchmark and identify practical leakage across models, data distributions, and adaptation types. We see our findings as a foundation for future theoretical work, and hope the benchmark will enable further analysis along these lines.
>
> We also include discussion points that may guide theoretical follow-ups. In particular, we highlight the connection between the number of parameters of the adaptation methods and the privacy risk.
>
> >**Including the error bar could make the empirical results and conclusions more convincing**
>
> We understand that including error bars everywhere would strengthen our empirical results and conclusions. However, conducting a proper statistical analysis would require us to adapt over 500 models, multiplied by the number of seeds, for each individual LLM run, given that our benchmark comprises  7 models across 4 adaptations on 6 different datasets, and 3 epsilon values. Thus, even with only 3 seeds, this would account for more than 1500 models for a single hyperparameter configuration, thereby requiring a prohibitive amount of compute.
>
> To address the Reviewer’s comment and increase the statistical rigour of our results, we were able to run the main evaluations (corresponding to Tables 1 and 2 in the main paper) over 3 different seeds. We report the mean and standard deviation below. The main conclusions are consistent with the results reported in our paper.
>
> |Dataset|$\varepsilon$|FT||LoRA||
> |----------------|-----|------------|------------|------------|------------|
> |||PPL|AUC|PPL|AUC|
> |**GermanWiki**|0.1|15.37|0.53$\pm $0.004|15.12|0.59$\pm $0.009|
> ||8|14.65|0.78$\pm $0.005|14.54|0.69$\pm $0.006|
> |**BookCorpus2 (train)**|0.1|20.34|0.63$\pm $0.04|19.98|0.51$\pm $0.009|
> ||8|19.83|0.75$\pm $0.006|19.83|0.73$\pm $0.01|
> |**Enron (Val)**|0.1|11.13|0.66$\pm $0.006|11.02|0.52$\pm $0.009|
> ||8|11.02|0.79$\pm $0.007|10.98|0.77$\pm $0.001|
>
>
>
> >**The evaluation focuses primarily on the Pythia family and GPT-Neo models. Including more diverse architectures would be great (e.g. gemma).**
>
> We focus on Pythia and GPT-Neo because these models were trained on the PILE dataset, which is openly available and has a validation set. This provides us with complete knowledge of the pre-training dataset (for the overlap experiments), along with IID data from the validation set, and the ability to choose OOD data that is far from the training distribution. When using models like Gemma, which are only open-weight but not fully open-source, we cannot confidently assess which data is overlapping, IID, and OOD.
>
> To address the Reviewer’s comment, we conducted additional experiments on two recent fully open models from a different family, namely, **Olmo-1b** and **Olmo-2-0425-1b**. Since these models rely on the Dolma and Dolmino datasets, which lack available validation sets,  we can only confidently report the results for the overlap and OOD experiments.
>
> For the overall experimental setup, we followed the main paper and reported our findings for LoRA and full-finetuning (FT). We include both perplexity (PPL) and AUC from RMIA with one shadow model and use Dolma’s Wiki train as overlap data and the external GermanWiki as OOD data.
>
>
> **Olmo-1B-hf (Dolma (Wiki))**
> |$\varepsilon$| FT PPL| FT AUC|LoRA PPL|LoRA AUC|
> |-|-|-|-|-|
> |0.1|17.98|0.5|15.53|0.5|
> |8|15.96|0.62|14.60|0.82|
> |$\infty$|14.78|1|14.27|0.85|
>
> **Olmo-1B-hf (GermanWiki)**
> |$\varepsilon$| FT PPL|FT AUC|LoRA PPL|LoRA AUC|
> |-|-|-|-|-|
> |0.1|22.78 |0.52|21.05|0.67|
> |8 |20.86 |0.65|18.88|0.69|
> |$\infty$|16.365|1|17.34|0.84|
>
>
> **OLMo-2-0425-1B (Dolmino (Wiki))**
> |$\varepsilon$| FT PPL|FT AUC|LoRA PPL|LoRA AUC|
> |-|-|-|-|-|
> |0.1|12.73|0.5|12.87|0.61|
> |8|12.58|0.778|12.72|0.81|
> |$\infty$|12.18|0.932|12.10|1|
>
> **OLMo-2-0425-1B (GermanWiki)**
> |$\varepsilon$| FT PPL|FT AUC|LoRA PPL|LoRA AUC|
> |-|-|-|-|-|
> |0.1|22.45|0.52|20.44|0.56|
> |8|20.37|0.79|19.62|0.72|
> |$\infty$|17.47|1|19.02|0.92|
>
> The results align with the findings from other model families as discussed in detail in the paper. Particularly, when the adaptation data distribution is close to the pretraining data, the privacy risk increases.

---

> ### Comment · Reviewer_MdW3 · 2025-08-03
>
> Thank the authors for their effort and response. My questions are adequately answered, and I appreciate the authors for offering the off-the-shelf tool in their new pipeline. Therefore, I will keep my current rating.
>
> I also read through other reviewers' comments, but found myself not very familiar with this area and had some bias. Therefore, I have lowered my confidence to be responsible.

---

> > ### Author Response · Authors · 2025-08-04
> > **Thank you for maintainig the high rataing and we are glad that our answers addressed the concerns**
> >
> > Dear Reviewer MdW3,
> >
> > Thank you very much for the interaction and maintaining the positive score. We are happy that Reviewer's questions were adequately answered. Shall there be any further questions or concerns, we remain available to answer.
> >
> > With kind regards,
> >
> > Authors of the submission #256

---

### Official Review · Reviewer_xVUq · 2025-07-03

**Rating:** 4
**Confidence:** 2

**Summary:**

This paper investigates the practical privacy leakage risks in the pretrain–adapt paradigm, focusing on the use of differential privacy (DP) when adapting large language models (LLMs). The authors systematically benchmark how the distributional relationship between adaptation data and pretraining data affects empirical leakage, highlighting that greater overlap between the two amplifies the risk. This study addresses an important gap between theoretical DP guarantees and real-world privacy risks, offering actionable insights and a structured auditing framework for deploying LLMs in sensitive domains.

**Dataset Code Accessibility:**

Partly

**Ethical Considerations:**

No, there are no or only very minor ethics concerns

**Final Justification:**

Thank the authors for the clarifications, and I will keep my original rating. However, as I am not very familiar with this specific area, I would suggest that the AC place more weight on the opinions of reviewers with higher confidence scores.

**Limitations Weaknesses:**

- The paper does not propose any new DP mechanism or attack technique; instead, it mainly combines strong existing attack methods with systematic benchmarking. As a result, the theoretical contribution is somewhat limited;

- The discussion of the utility–privacy trade-off is relatively brief and lacks concrete guidance on how to balance privacy protection and model performance in real-world deployments;

- Compared to related work such as PrivAuditor and LLM-PBE, the key differentiator of this study is its focus on the impact of data distribution overlap, rather than the development of fundamentally new privacy-enhancing techniques.

**Strengths Contributions:**

- This work addresses a clear and relevant gap: although DP-based fine-tuning of LLMs is crucial for sensitive applications such as healthcare and law, there is a lack of systematic empirical evidence on its practical effectiveness and risks.

- The study thoroughly explores a wide range of adaptation methods, data distributions, model sizes, and attacker assumptions, providing conclusions with strong external validity and practical relevance.

- The paper is clearly written, with well-motivated research questions, rigorous experiments, and insightful discussions that will benefit both researchers and practitioners.

---

> ### Author Rebuttal · Authors · 2025-07-31
>
> We appreciate the Reviewer's positive feedback and for acknowledging our benchmark as bridging a relevant gap through our “well-motivated research questions, rigorous experiments, and insightful discussions.”
>
> >**The paper does not propose any new DP mechanism or attack technique; instead, it mainly combines strong existing attack methods with systematic benchmarking. As a result, the theoretical contribution is somewhat limited;**
>
> We agree that our work does not propose a new DP mechanism or attack. This is by design: our submission targets the **Benchmark and Dataset track**, which values systematic, reproducible evaluations over novel algorithms.
>
> Our contribution lies in providing a systematic benchmark to address the gap in understanding the **empirical privacy risks of DP adaptations**. Therefore, we provide an open-source code framework and answer five fundamental research questions that yield new insights into the privacy of LLM adaptations and assess real-world privacy risks of DP adaptations in various scenarios.
>
> We note that, in Section 6 of our submission, we formalize the framework for a holistic privacy auditing of LLMs. In total, we identify four different stages of auditing that need to be considered
> (see Figure 4) under the pretrain-adapt paradigm. We model the privacy audits as an adversarial game, where the main task is to guess if a given data point x was in a model’s training set or not.
>
> >**The discussion of the utility–privacy trade-off is relatively brief and lacks concrete guidance on how to balance privacy protection and model performance in real-world deployments;**
>
> While we agree that discussing privacy-utility trade-offs is relevant, we also note that this trade-off is highly use-case dependent. For example, adapting an LLM on private medical data requires choosing a different trade-off than adapting the model on public data. Our customizable benchmark code allows practitioners to faithfully assess the trade-offs and leakage in their own setting, with their own data, and to opt for privacy guarantees that align with their desired practical protection. We have added a discussion on this to the appendix and extended the README in our codebase to make such evaluations as accessible as possible.
>
> While the guidance on trade-offs is individual, our benchmark provides general guidance based on our general findings, including opting for adaptations, such as LoRA, that achieve the highest empirical protection, and considering their data distribution w.r.t. to the pretrained model.
>
> >**Compared to related work such as PrivAuditor and LLM-PBE, the key differentiator of this study is its focus on the impact of data distribution overlap, rather than the development of fundamentally new privacy-enhancing techniques.**
>
> Our focus is deliberately on auditing and evaluating privacy, not on proposing new mechanisms. The main difference between our benchmark and PrivAuditor is that we evaluate DP adaptations, whereas they limit their scope to non-private scenarios (covered in our benchmark by epsilon = infinity). LLM-PBE systematizes privacy risks across the full model lifecycle, but lacks extensive auditing of DP-adaptations (e.g., lack of PEFT methods) as well as understanding of interdependencies between data (IID, overlap scenarios). Beyond overlap, our work is also the first to provide a thorough benchmark of practical privacy leakage under DP’s theoretical privacy guarantees for LLM adaptations.
>
> We hope that our responses help clarify the points raised, and would be grateful if the Reviewer might consider updating their score accordingly.

---

> > ### Comment · Reviewer_xVUq · 2025-08-05
> >
> > Thank the authors for the clarifications, and I will keep my original rating. However, as I am not very familiar with this specific area, I would suggest that the AC place more weight on the opinions of reviewers with higher confidence scores.

---

### Official Review · Reviewer_51CB · 2025-07-04

**Rating:** 4
**Confidence:** 2

**Summary:**

Large Language Models (LLMs) are increasingly adapted for sensitive domains (e.g., medicine), often using Differential Privacy (DP) for protection. However, the real-world privacy effectiveness of DP adaptations remains unclear, especially due to potential overlaps or similarities between pretraining and adaptation data.

**Dataset Code Accessibility:**

Yes

**Ethical Considerations:**

No, there are no or only very minor ethics concerns

**Final Justification:**

The authors have well addressed my concerns. I will keep my positive score.

**Limitations Weaknesses:**

Experiments are mainly performed on the Pythia and GPT-Neo families, which may limit generalizability to more advanced or proprietary LLMs (e.g., GPT-4, Gemini).

Even under moderate DP budgets (e.g., ε = 8), leakage remains significant, raising concerns about DP's sufficiency without complementary safeguards.

The strongest privacy attacks assume access to similar shadow models, which may not always reflect real-world scenarios.

**Strengths Contributions:**

The paper rigorously evaluates privacy risks in differentially private (DP) LLM adaptations using robust attacks (e.g., RMIA, canary extraction), covering various adaptation methods, datasets (IID, OOD, overlapping), and privacy budgets.

It reveals that data distribution similarity—rather than exact overlap—between pretraining and adaptation sets strongly affects privacy leakage.

LoRA and Prefix Tuning are shown to provide relatively better empirical privacy, especially for OOD data.

---

> ### Author Rebuttal · Authors · 2025-07-31
>
> We thank the Reviewer for their positive feedback and for acknowledging our *rigorous* evaluation and various interesting findings on data distribution and adaptation methods.
>
> >**Experiments are mainly performed on the Pythia and GPT-Neo families, which may limit generalizability to more advanced or proprietary LLMs (e.g., GPT-4, Gemini).**
>
> Our analysis cannot be conducted on GPT4 or Gemini, since these are **closed** models, i.e., not publicly released. As a consequence, 1) their APIs do not offer the gradient-based adaptation capabilities with differential privacy; 2) we cannot access information about their training datasets, making it impossible to identify IID and OOD data for our analyses; and 3) it is not possible to perform expressive MIAs since the models do not output detailed token-probabilities/logits.
>
> Yet, to strengthen our findings and address the Reviewer’s concerns, we expanded our evaluation to include other recent open models. Therefore, we ran additional experiments on **OLMo-1B-hf** and **OLMo-2-0425 1B**. Since these models were trained on Dolma and Dolmino, respectively, and neither of them provides a validation set, our experiments are limited to *overlap* (using the known train data) and *OOD* scenarios using GermanWiki, as OLMo is trained on mainly English documents. Below, we report results for both full fine-tuning (FT) and LoRA.
>
>
> **OLMo-1B-hf - Dolma (Wiki)**
> |$\varepsilon$| FT PPL| FT AUC|LoRA PPL|LoRA AUC|
> |-|-|-|-|-|
> |0.1|17.98|0.5|15.53|0.5|
> |8|15.96|0.62|14.60|0.82|
> |$\infty$|14.78|1|14.27|0.85|
>
> **OLMo-1B-hf - GermanWiki**
> |$\varepsilon$| FT PPL|FT AUC|LoRA PPL|LoRA AUC|
> |-|-|-|-|-|
> |0.1|22.78 |0.52|21.05|0.67|
> |8 |20.86 |0.65|18.88|0.69|
> |$\infty$|16.365|1|17.34|0.84|
>
>
> **OLMo-2-0425-1B Dolmino (Wiki)**
> |$\varepsilon$| FT PPL|FT AUC|LoRA PPL|LoRA AUC|
> |-|-|-|-|-|
> |0.1|12.73|0.5|12.87|0.61|
> |8|12.58|0.778|12.72|0.81|
> |$\infty$|12.18|0.932|12.10|1|
>
> **OLMo-2-0425-1B GermanWiki**
> |$\varepsilon$| FT PPL|FT AUC|LoRA PPL|LoRA AUC|
> |-|-|-|-|-|
> |0.1|22.45|0.52|20.44|0.56|
> |8|20.37|0.79|19.62|0.72|
> |$\infty$|17.47|1|19.02|0.92|
>
> Our results show consistent trends with the ones we reported in the paper for other models, especially when the adaptation data distribution is close to the pretraining data. The privacy risks increase when the adaptation data is closer to the pretraining distribution.
>
> >**Even under moderate DP budgets (e.g., $\varepsilon$ = 8), leakage remains significant, raising concerns about DP's sufficiency without complementary safeguards.**
>
> We fully agree with the Reviewer, and this is, in fact, one of the central takeaways of our work. Our findings underscore the need for practitioners to rely on both tighter privacy budgets (lower $\varepsilon$) and adaptation methods that provide stronger empirical privacy protection, such as LoRA. Our benchmark is specifically designed to surface these practical trade-offs.
>
> >**The strongest privacy attacks assume access to similar shadow models, which may not always reflect real-world scenarios.**
>
> In our experiments, we comprehensively cover different attacker knowledge scenarios. Especially, we also assess shadow models that are not similar to the target model in Appendix D. While those separate shadow models achieve a lower attack success in comparison to using the original pretrained model, they still detect residual privacy leakage while representing more realistic scenarios. Our full MIA analysis in Appendix C.1 covers the attack success rates for various setups (*same* vs. *different* vs. *no* shadow model attacks), and hence, provides a comprehensive insight into the spectrum of leakage.
>
> ---
> We hope that our responses help clarify the points raised, and we would be grateful if the Reviewer might consider updating their rating accordingly.

---

> > ### Comment · Reviewer_51CB · 2025-08-05
> > **keep my original socre**
> >
> > Thank the authors for the clarifications, and I will maintain my positive socre.

---

### Official Review · Reviewer_Ja7d · 2025-07-23

**Rating:** 5
**Confidence:** 3

**Summary:**

The authors present a large-scale empirical study on privacy attacks on adapting LLMs to private data. They perform privacy auditing on the result of adaptation, where they:
vary the attack;
vary the adaptation method;
vary the base model; and
vary the adaptation data.

They also lay out a taxonomy of the new auditing games one may be interested in in the pretrain-finetuning paradigm.

**Additional Feedback:**

I very much appreciate this paper, however my concerns with the technical evaluation make it difficult for me to recommend acceptance.

If authors could address my concerns about the experimental study design and lack of utility metrics, I am very happy to raise the score.

**Dataset Code Accessibility:**

Yes

**Dataset Code Comments:**

The code looks usable, however it feels quite specific to this study and there is not much elaboration on how one would use it as an artifact.

**Ethical Considerations:**

No, there are no or only very minor ethics concerns

**Final Justification:**

The authors addressed my concerns regarding the choice of a fixed perplexity of evaluating different lora hyperparameters, by presenting the full perplexity-auc curve. They also introduced utility numbers and demonstrated revisions to their code to improve its usability as an artifact for the community. Therefore I recommend Accept.

From their rebuttal, we do see that the picture is less clear -- different hyperparameters for a fixed method can lead to significantly different attack success rates. I lowered my confidence due to the lack of clear description of the hyperparameter tuning procedure and justification that their choices are sensible (e.g. comparison to external baselines).

**Limitations Weaknesses:**

For the main experiment comparing different types of adaptation methods and data, I have disagreements with the empirical design of the study.

The question of interest to a practitioner is the following: for a given private dataset, model, and adaptation method, if I use it, what kind of privacy risks should I expect?

However “using it” implies selecting a set of hyperparameters representative for the dataset, model, and setting. Results in Empirical Privacy Variance (Hu et al., 2025) demonstrate that different hyperparameters for the same algorithm at the same privacy budget can affect auditing results. Hence, I think the only way to proceed is to pick the best hyperparameters for the adaptation method, and see where methods lie on the utility/empirical privacy frontier (similar to Figure 8 in the appendix).

Instead, authors remark that: “Since membership inference success is highly dependent on the train-test gap, for a fair comparison of the privacy leakage, we ensure similar evaluation perplexities, in particular, similar validation loss values at the end of the adaptation’s training for specific datasets across adaptation methods”.

LoRA may leak less at the particular test loss considered by the study design, but (1) there is little discussion on what this value is and how its picked, and whether results are robust to the choice; and (2) Potentially in practice, LoRA could result in more leakage if it can be finetuned to lower test loss at the same privacy budget.

Furthermore, of interest to the practitioner is if these conclusions are in utility regimes that are interesting, however this cannot be assessed from the current manuscript as utility is not reported for these adapted models.

Secondly, While the results are a valuable empirical study, the artifacts produced (benchmark) feel quite specific to their study. It would be great if the authors could elaborate on the contribution of their work as an artifact to the community instead of simply a set of results in a paper.

**Strengths Contributions:**

The paper addresses an important gap in the literature: the bulk of MIA work studies from-scratch training, yet most practical applications of DP-SGD use finetuning.

There are lots of results and they are presented in a very comprehensible manner.

There are quite a few interesting findings: (1) not much difference between in distribution data and directly overlapping data; (2) the extent to which access to the base checkpoint improves MIA results; (3) the difference in adaptation methods at the same privacy budget, evaluated at similar utility.

The RMIA baseline is sufficiently powered, demonstrating perfect AUC across all tasks and datasets for the 1B model at epsilon=infty (with the sole exception of LoRA tuning on one of the OOD datasets), and high scores at epsilon=8.

Authors present a large amount of experimental settings that allow one to make conclusions about general trends robustly. For instance, using multiple privacy attacks (RMIA and Reference) and observing correlated results, also testing with multiple id and ood sets.

---

> ### Author Rebuttal · Authors · 2025-07-31
>
> We thank the Reviewer for the constructive feedback. We appreciate the Reviewer’s recognition that our work addresses *an important gap* in the literature, presents the results in a *comprehensive* manner, and yields interesting findings.
>
> >**Study Design**
>
> We focus on comparing different adaptation methods under controlled evaluation perplexities to decouple privacy leakage from utility-induced behavior. Varying train-test gaps may otherwise result in a confounding factor that can affect MIA success (the higher the gap, the better the performance of a MIA). Our design allows us to attribute the observed leakage differences more directly to the adaptation method itself. In addition, we examine diverse potential dataset relationships between private adaptation and pretraining datasets.  Moreover, to limit potential biases, we focus on a range of models trained on the same corpora.
>
> >**The question of interest to a practitioner is the following: for a given private dataset, model, and adaptation method, if I use it, what kind of privacy risks should I expect?**
>
> We do agree with the Reviewer that practitioners are interested in the privacy-utility trade-offs. Therefore, we ran new experiments to provide a privacy-utility trade-off curve to address this perspective. We also added the curves to Appendix C in the paper. Given the format of the rebuttal, we present our results in a table below.
>
> **Experimental Setup**:
>
> Concretely, we vary the hyperparameters in a grid search and report the resulting trade-offs between privacy and utility for Pythia 1B at $\varepsilon=8$. We report here the four best runs for the given adaptation method, dataset, and privacy budget.
>
> Our findings suggest that hyperparameter choices can lead to varying levels of utility and privacy risks. Overall, we find that LoRA offers the best privacy-utility trade-offs, i.e., at the same utility level (measured by perplexity (PPL)), it exhibits lower privacy risks (as indicated by the AUC value for the RMIA with one shadow model). For example, for GitHub (val), at perplexity 4.8, LoRA exhibits an AUC of 0.6, whereas FT at the same perplexity has 0.83. These findings align with the trends reported in our main paper.
>
> In addition to the results presented here, we also created privacy-utility curves for the Pythia-1B model adapted to GermanWiki and BookCorpus2 (both validation and training sets) across all three privacy levels, and all four adaptation methods: Prefix, LoRA, Full fine-tuning (FT), and Head fine-tuning (HT). We included those findings in Appendix C.
>
>
> **BookCorpus (val)**
> |Adaptation|PPL|AUC|
> |-|-|-|
> |LoRA|20.21|0.52|
> |LoRA|20.08|0.72|
> |LoRA|20.07|0.73|
> |LoRA|20.03|0.73|
> |LoRA|19.94|0.75|
> |LoRA|19.85|0.78|
> |FT|20.11|0.78|
> |FT|20.1|0.81|
> |FT|20.05|0.82|
> |FT|19.93|0.82|
> |HT|20.1|0.72|
> |HT|20.11|0.77|
> |HT|20.11|0.8|
> |HT|20.07|0.87|
> |Prefix|19.92|0.89|
>
>
>
> **Github (val)**
> |Adaptation|PPL|AUC|
> |-|-|-|
> |FT|4.79|0.81|
> |FT|4.8|0.83|
> |FT|4.77|0.86|
> |FT|4.8|0.87|
> |LoRA|4.78|0.53|
> |LoRA|4.79|0.76|
> |LoRA|4.8|0.6|
> |LoRA|4.72|0.84|
> |HT|4.92|0.54|
> |HT|4.88|0.65|
> |HT|4.78|0.95|
> |HT|4.80|0.98|
> |Prefix|4.79|0.93|
>
>
> **GermanWiki**
> |Adaptation|PPL|AUC|
> |-|-|-|
> |LoRA|14.54|0.679|
> |LoRA|14.59|0.671|
> |LoRA|14.43|0.71|
> |LoRA|15.301|0.571|
> |FT|15.033|0.538|
> |FT|15.068|0.652|
> |FT||14.74|0.69|
> |FT|14.85|0.79|
> |HT|14.7|0.82|
> |HT|14.96|0.93|
> |HT|15.31|0.65|
> |HT|15.01|0.84|
> |Prefix|25.96|0.57|
>
>
> >**utility is not reported for these adapted models.**
>
> We reported the loss values as a universal proxy for all our experiments in Appendix E, Tables 19-26 (in our original submission).
>
> We do agree with the Reviewer that the utility might be more intuitively interpretable for practitioners. Therefore, in the table below, we present the Rouge-1 score (R1) and perplexity (PPL) for Pythia-1b when adapted for the SAMSum dataset below.
> |Adaptation $\downarrow$|$\varepsilon=\infty$||$\varepsilon=8$||$\varepsilon=0.1$||
> |-|-|-|-|-|-|-|
> |Metric $\rightarrow$ |PPL|R1|PPL|R1|PPL|R1|
> |Prefix|8.306|43.09|8.895|35.61|14.804|12.32|
> |LoRA|8.279|44.00|8.806|38.44|10.890|17.88|
> |FT|8.070|44.33|8.794|38.81|13.882|21.47|
> |HT|9.584|31.25|10.098|26.51|14.708|15.14|
>
>
> Since lower loss values correspond directly to higher utility, the loss (that we reported) serves as an effective and generalizable proxy for evaluating utility.
>
> >**contribution of the work as an artifact to the community**
>
> Our benchmark contributes an open-source code suite for the community, including:
> 1. Standardized private and non-private LLM adaptation pipelines for multiple methods, including LoRA, Prefix Tuning, Full Fine Tuning, and Head Fine Tuning;
> 2. Joint implementation of privacy attacks to audit leakage;
> 3. Scripts to harmonize model evaluation and leakage reporting. Specifically, we integrated within an unified pipeline,  which directly compares privacy risks, utility, and trade-offs across various adaptations and privacy budgets.
>
> Our benchmark also provides a **systematic study with five research questions** that yield **novel insights** into the impact of data distributions, risks, and assumptions, and privacy of the adaptation methods, as acknowledged by the Reviewer. This aligns closely with the Benchmark and Dataset Track’s scope, which explicitly invites *“systematic analyses of existing systems on novel datasets yielding important new insight,”*  in the call for papers.  Our findings would be difficult to obtain without the controlled evaluation framework that we developed.
>
> Finally, in Section 6, we introduce a new **holistic framework** for privacy auditing that enables systematic benchmarking of privacy leakage in foundation models’ new pretrain-adapt training paradigm and can be leveraged by the community beyond our study.
>
> > **The code looks usable, however, it feels quite specific to this study, and there is not much elaboration on how one would use it as an artifact.**
>
> Following the Reviewer’s comment, to enhance the usability and generalizability of our framework, we added a unified pipeline that allows us to run multiple adaptations with various hyperparameters. Specifically, the new pipeline accepts .yaml files for defining and executing diverse adaptation configurations with varying hyperparameters. **Our new pipeline's design allows for the direct comparison of privacy risks, utility, and privacy-utility trade-offs for various privacy budgets and adaptations, and can serve the community as a valuable off-the-shelf tool.**
>
> We also improved our README file with clear instructions on adding custom datasets and integrating new adaptations, which makes the framework more accessible. For practitioners, this means a more streamlined approach to evaluating privacy-preserving adaptation for LLMs. Our framework may help them quickly assess various configurations, efficiently explore, and understand the trade-offs and potential risks.
>
> Following the submission guidelines, which disallow adding links, we will make our updated repository available in the camera-ready version, if accepted.
>
> ---
> If the above responses address some of the Reviewer's concerns, we would greatly appreciate it if they would consider raising their rating.

---

> > ### Comment · Reviewer_Ja7d · 2025-08-06
> > **Thank you for addressing my concerns**
> >
> > Thanks to the authors for providing the additional results. In particular the results showcasing the tradeoff between PPL and AUC for the different adaptation methods which resolves my concerns.
> >
> > I plotted the figures provided by the authors and can confirm the PPL-AUC pareto frontier is dominated by lora, echoing the results of the paper than LoRA is generally safer when controlling for utility.
> >
> > I am concerned a bit with the hyperparameter ranges used for training -- in particular sweeps using batch size 1-64 which are quite far from optimal choice of large batches in DP. I acknowledge that compute limitations are likely the cause of these choices.
> >
> > In the final version of the text, I hope the authors could keep in mind the concerns regarding evaluating the adaptations at a fixed perplexity, and explain in detail their methodology for choosing a particular perplexity.
> >
> > Thank you for the utility results. Pointing to external baselines and or few-shot results with the base model would better contextualize the results.
> >
> > I trust that the final version of the code will be improved with the new changes regarding usability.
> >
> > I raised my score. However I have also lowered my confidence. The attack AUC results do seem to show some strong variations based on the the hyperparameters, which merits a more careful explanation and justification of the hyperparameter search strategy.

---

> > > ### Author Response · Authors · 2025-08-07
> > > **Thank You for Increasing the Score and Your Interest in Our Results**
> > >
> > > Dear Reviewer Ja7d,
> > >
> > > We appreciate your thorough engagement with our work and the time you invested in plotting and analyzing our additional results. Your insights about the hyperparameter selection are particularly valuable. We acknowledge that our choices were constrained by available computational resources and will discuss these limitations in the final version.
> > >
> > > We are grateful for your engagement and the score adjustment.
> > >
> > > With kind regards,
> > >
> > > Authors of the submission #256

---

> ### Comment · Area_Chair_D4vM · 2025-08-06
> **Urgent Reminder: Reviewer Discussion**
>
> Dear Ja7d,
>
> Thank you for support to NeurIPS! This is a reminder that Reviewer-Author discussions is extended by 48h till Aug 8, 11.59pm AoE.
>
> Since you have not participated in the discussion, I want to reiterate the following key points:
>
> 1. Reviewers are expected to stay engaged in discussions,
> - It is not OK to stay quiet.
> - It is not OK to leave discussions till the last moment.
> - If authors have resolved your (rebuttal) questions, do tell them so.
> - If authors have not resolved your (rebuttal) questions, do tell them so too.
>
> 2.	Please note “Mandatory Acknowledgement” button is to be submitted only when reviewers fulfill all conditions below (conditions in the acknowledgment form):
> - Read the author rebuttal
> - Engage in discussions (reviewers must talk to authors, and optionally to other reviewers and AC - ask questions, listen to answers, and respond to authors)
> - Fill in "Final Justification" text box and update “Rating” accordingly (this can be done upon convergence - reviewer must communicate with authors first)
>
> Finally, please treat authors the way you would like to be treated (fairness, politeness, calmness, attention and focus on merits). Thanks again for your time and effort.
>
> Best regards,
>
> AC

---

### Comment · Area_Chair_D4vM · 2025-08-03
**Reminder for Reviewer-Author discussions**

Dear reviewers,

Thank you so much for all your time and effort supporting NeurIPS!

If you haven't yet, please take a moment to read through the author's rebuttal. The reviewer-author discussion period is crucial for ensuring a fair and comprehensive evaluation of their work. If the rebuttal addresses your concerns, please acknowledge this and adjust your scores accordingly. If not, please let them know which concerns remain and if you have any follow-up questions. Your thoughtful feedback will help authors improve their scholarship and propel our field forward.

I know this is a busy time, and really appreciate your effort.



Best Regards
Area Chair

---

### Decision · Program_Chairs · 2025-09-18

**Decision:**

Reject

**Comment:**

This paper works on the benchmarking empirical privacy protection of DP LLMs within the pretrain-adapt paradigm. It shows privacy auditing: vary two attacks(robust membership inference, and canary data extraction.); two data distributions (exact overlaps with pretraining data through IID, and OOD examples); four adaptation methods (full DP fine-tuning , last layer DP fine-tuning, parameter-efficient fine-tuning (PEFT) methods, DP-Prefix Tuning,); two families of base models (Pythia #, GPT Neo # ); and six datasets.

Findings:

1.	The distributional closeness between pretraining and adaptation data is a key factor influencing empirical privacy leakage.
2.	Not all DP adaptation methods offer equal protection, even when enforcing the same formal level guarantee. PEFT methods, such as LoRA, offering significantly lower privacy risks while maintaining strong utility.
3.	Highlight the need for stringent DP constraints (e.g., ε < 0.1) to mitigate privacy risks in LLM adaptations effectively
4.	Attackers gain a substantial advantage when they have access to the original pretrained LLM used during adaptation.
5.	Suggest that privacy assessments should not treat pretraining and adaptation in isolation.

Strengths:

1. This paper is overall well-written with clear structure, informative figures and tables.
2. The benchmark demonstrates exceptional thoroughness, systematically varying critical factors including data distribution relationships (overlap, IID, OOD), adaptation methods, privacy regimes, and model architectures.
3. The work addresses a critical gap between theoretical DP guarantees and empirical privacy protection in real-world LLM deployment scenarios.

Weakness:

The evaluation focuses on open models, such as the Pythia family and GPT-Neo models. Analysis cannot be conducted on other important but closed models such as GPT4 or Gemini.

Discussions:

Reviewers mainly concern about the experiments design, such as less of utility, and less of LLMs, and different hyperparamters. Authors answers about that.

===== FINAL UPDATE FROM DB Track PCs ====

The final decision for this paper has been taken by the program chairs after consultation with the SACs. All Senior Area Chairs have ranked papers according to the feedback from the AC during the review process. We decided to leave the original meta-review to reflect the opinion of the AC in light of the initial discussions with reviewers and SAC.